# How well did experts and laypeople forecast the size of the COVID-19 pandemic?

**Gabriel Recchia** *, **Alexandra L. J. Freeman** , **David Spiegelhalter**

Department of Pure Mathematics and Mathematical Statistics, Winton Centre for Risk and Evidence Communication, University of Cambridge, Cambridge, United Kingdom

* glr29@cam.ac.uk

## Abstract

Throughout the COVID-19 pandemic, social and traditional media have disseminated predictions from experts and nonexperts about its expected magnitude. How accurate were the predictions of 'experts'—individuals holding occupations or roles in subject-relevant fields, such as epidemiologists and statisticians—compared with those of the public? We conducted a survey in April 2020 of 140 UK experts and 2,086 UK laypersons; all were asked to make four quantitative predictions about the impact of COVID-19 by 31 Dec 2020. In addition to soliciting point estimates, we asked participants for lower and higher bounds of a range that they felt had a 75% chance of containing the true answer. Experts exhibited greater accuracy and calibration than laypersons, even when restricting the comparison to a subset of laypersons who scored in the top quartile on a numeracy test. Even so, experts substantially underestimated the ultimate extent of the pandemic, and the mean number of predictions for which the expert intervals contained the actual outcome was only 1.8 (out of 4), suggesting that experts should consider broadening the range of scenarios they consider plausible. Predictions of the public were even more inaccurate and poorly calibrated, suggesting that an important role remains for expert predictions as long as experts acknowledge their uncertainty.

## Introduction

Expert opinion is undoubtedly important in informing and advising those making individual and policy-level decisions. In the early COVID-19 pandemic, clinicians, epidemiologists, statisticians, and other individuals seen as experts by the media and the general public, were frequently asked to give off-the-cuff answers to questions about how bad the pandemic might get. Answers to such questions draw upon "skilled intuition," i.e., rapid judgments based on the recognition of similarities to other relevant situations, built up over a long period of experience [1, 2]—as well as the efficient recall of relevant information from long-term memory, which also benefits from expertise [3]. However, as the quality of expert intuition can vary drastically depending on the field of expertise and the type of judgment required [2], it is important to conduct domain-specific research to establish how good expert predictions really are, particularly in cases where they have the potential to shape public opinion or government policy.

**Data Availability Statement:** Data and code is available in the OSF project "How well did experts and laypeople forecast the size of the COVID-19 pandemic?", available at https://osf.io/dcn5q.

**Funding:** This work was funded by the Winton Centre for Risk and Evidence Communication (https://wintoncentre.maths.cam.ac.uk/), which is supported by a donation from the David and Claudia Harding Foundation. There is no associated grant number. The funders had no role in study design, data collection and analysis, decision to publish, or preparation of the manuscript.

**Competing interests:** The authors have declared that no competing interests exist.

There has been limited research on the accuracy of expert and nonexpert COVID-19 forecasts, or the accuracy of levels of confidence in such forecasts ("calibration"). One study of COVID-19-related predictions from 41 U.S. experts found that their proposed method of aggregating predictions into a combined consensus distribution was more accurate than a more naïve way of aggregating expert forecasts which they referred to as an "unskilled forecaster" [4], but did not actually survey nonexperts or evaluate nonexpert predictions. Conversely, [5] investigated COVID-19 forecasts of nonexperts in Germany in the early pandemic, but did not investigate expert forecasts. One notable finding was that in a survey conducted in mid-March that asked participants to predict how many COVID-19 deaths would have occurred in Germany by the end of the year, the median estimate was exceeded just 16 days after the survey. We are not aware of any research specifically comparing expert and non-expert COVID-19 forecasts, although COVID-19 statistical and computational models and some of the expert forecasts based on them have been roundly critiqued for inaccuracy, overconfidence, and flawed assumptions [6–8].

It is important to differentiate between research evaluating the forecasts of 'experts'—operationally defined in this paper as individuals holding occupations or roles in subject-relevant fields, such as epidemiologists and statisticians—and research evaluating specific epidemiological models, although expert forecasts may well be informed by epidemiological models. There has been more research evaluating the latter, which has found that many COVID-19 models achieve reasonable short-term predictions but that longer-term predictions are far more difficult due to the nonlinear nature of the processes that drive the spread of infection [9–11]. It is also important to distinguish between forecasts of arbitrary 'experts' as defined above, and those of the much smaller subset of experts who most directly inform public policy (e.g., scientific advisory committees, scientists participating in science policy initiatives, etc.); the latter are far harder to study, and we necessarily focus on the former here. Arguably, even the forecast accuracy of 'experts' by the broad definition is of some interest—should we put any more weight on an epidemiologist's Twitter forecasts, or on those of someone interviewed by local media because they are a well-known statistician, than the predictions of a man or woman stopped on the street? In any case, systematic analysis of predictions for other viral outbreaks provides hints that we perhaps should not expect too much from either expert or nonexpert forecasts of COVID-19. For example, research conducted on disease forecasts (expert forecasts as well as model predictions) of the 2014 Ebola outbreak appearing in the published literature found that only 37% of predictions ended up being within 50% - 150% of the actual number of deaths [12]; furthermore, of seven predictions that were considered to represent "best case scenarios", four of them still predicted death tolls higher than what ultimately resulted. As with the literature focusing specifically on COVID-19, evaluations of forecasts for other viral outbreaks have largely focused more on proposing model evaluation frameworks or evaluating specific models (e.g. [13, 14]) than on expert forecasts.

However, to contextualize conclusions about expert predictions, it is critical to compare them to nonexpert predictions [15]. After all, if expert predictions are disregarded by the public, nonexpert predictions are liable to drive behavior in their stead. To this end, we conducted a survey of experts and nonexperts in April 2020, asking participants to make four predictions about the extent and severity of the COVID-19 outbreak by the end of 2020, and to indicate their confidence in their predictions by providing lower and upper bounds of an interval that they were 75% confident the true answer would fall within. (We refer to these as '75% confidence intervals', following prior similar literature; this usage is clearly distinct from the traditional notion of a 'confidence interval' as an estimate computed from the statistics of observed data). The results provided clear evidence of differences between expert and nonexpert predictions in both accuracy and calibration.

**Table 1. Questions asked of participants with corresponding forecast medians, median absolute deviation (MAD), median absolute error (MAE) and median relative error (MRE).**

|  | *Question 1* | *Question 2* | *Question 3* | *Question 4* |
|---|---|---|---|---|
| *Question* | How many people in the country you're living in do you think will have died from COVID-19 by December 31st 2020? | How many people in the country you're living in do you think will have been infected by COVID-19 by December 31st 2020? | Out of every 1000 people who will have been infected by the virus worldwide, how many do you think will have died by December 31st 2020 as a result? | Out of every 1000 people who will have been infected by the virus in the country you're living in, how many do you think will have died by December 31st 2020 as a result? |
| *How true outcome estimate was derived* | Total number of "deaths within 28 days of positive test" having a date of death earlier than 1 Jan 2021 | Number of infections implied by dividing the total number of COVID-19 deaths in the UK (left) by the UK infection fatality rate estimated by Imperial College COVID-19 response team in Oct 2020 | 1000 multiplied by the age-specific infection fatality rates estimated by the Imperial College COVID-19 response team in Oct 2020, weighted by worldwide age distribution | 1000 multiplied by the UK infection fatality rate estimated by the Imperial College COVID-19 response team in Oct 2020 |
| *True outcome estimate* | 75,346 | 6,385,254 | 4.55 | 11.8 |
| *Experts, median (MAD)* | 30,000 (15,000) | 4,000,000 (3,687,500) | 10 (5) | 9.5 (4.5) |
| *High-numeracy nonexperts, median (MAD)* | 25,000 (10,000) | 800,000 (700,000) | 30 (20) | 30 (22) |
| *All nonexperts, median (MAD)* | 20,000 (10,000) | 250,000 (247,000) | 50 (45) | 40 (35) |
| *Expert MAE* | 45,346 | 5,585,254 | 5.45 | 6.80 |
| *High-numeracy nonexpert MAE* | 55,346 | 6,085,254 | 25.45 | 18.20 |
| *Nonexpert MAE* | 55,346 | 6,235,254 | 45.45 | 28.20 |
| *Expert MRE* | 2.51 | 3.19 | 1.98 | 2.03 |
| *High-numeracy nonexpert MRE* | 3.32 | 7.98 | 5.59 | 3.20 |
| *Nonexpert MRE* | 3.77 | 25.54 | 9.19 | 3.98 |

## Materials and methods

Participants were asked to make four COVID-19 forecasts (Table 1). In each case they were asked to estimate what the true answer to the question would be, and afterward to provide two additional numbers "in such a way that you think there's about a 75% chance that the real-world answer will fall between your lower and higher number". 75% was considered a reasonable value as it made it possible to identify how well-calibrated participants were on average: it left room for identifying both overconfidence (if fewer than three of the four real-world outcomes fell within an individual's range) and underconfidence (if more than three did), and has been used in prior literature for similar reasons [16]. Nonexpert participants completed the survey as a part of a larger set of questions described elsewhere [17] which included the adaptive Berlin Numeracy test [18]. We surveyed 2,086 UK laypersons, sampling about half (N = 1047) from the survey platform Prolific Academic and the remainder from the ISO-certified panel provider Respondi.com, using proportional quota sampling to achieve a sample proportional to the UK population on age and gender. In parallel, we surveyed a convenience sample of experts recruited from social media. For the purposes of this survey, 'experts' were defined as epidemiologists, statisticians, mathematical modelers, virologists, and clinicians, as these represented the occupations/specialties of individuals commonly asked by the media to give predictions or expert opinions on COVID-19 in the early months of the pandemic. 140 respondents indicated that they resided in the UK and held one or more of these specialties.

All participants completed the survey between 7 April and 12 April 2020, with the exception of 5 experts who submitted responses between 14 and 16 April, one who submitted a response on 20 April, and one who submitted a response 7 May. The expert sample contained 19 epidemiologists, 65 statisticians, 44 mathematical modelers, 35 clinicians, and 1 virologist; this was defined as the list of "expert occupations" prior to survey distribution, and data from experts who did not hold one of these occupations/roles was excluded. (Numbers in the previous sentence do not sum to 140, as some individuals reported holding multiple roles). Answers to survey questions that were entered in non-machine-readable ways (e.g., the phrase "*1 million*" rather than "*1,000,000*") were normalized with a combination of regular expressions and manual inspection. Removing blanks, uninterpretable answers, answers exceeding 1000 (Q3 & Q4) or the population of the UK (Q1 & Q2), and answers which did not follow instructions yielded 405 point estimates and 402 ranges from experts, and 7,593 point estimates and 6,801 ranges from nonexperts.

Free text responses highlighted that some nonexpert participants found the instructions difficult to understand or follow, so we restricted our primary nonexpert analysis to those who scored in the top quartile of numeracy (N = 524), and reserved the full nonexpert pool for a secondary analysis. Accuracy and calibration were calculated by comparing participants' April estimates to a "true outcome estimate" determined in January 2021. Specifically, the total number of COVID-19 deaths by December 31 (Question 1) was assessed using the official criterion of the United Kingdom: the total number of "deaths within 28 days of positive test" having a date of death earlier than 1 Jan 2021, as reported at https://coronavirus.data.gov.uk/details/deaths [19]. The total number of infections by December 31 (Question 2) was estimated by computing the total number of COVID-19 deaths in the UK as of December 31 by the UK infection fatality rate (IFR) estimated by the Imperial College COVID-19 response team in October 2020 [20]. The true outcome estimate for Question 4 was obtained by multiplying this same estimated IFR by 1000. Finally, the true outcome estimate for Question 3 was obtained by multiplying the age-specific IFRs estimated by the Imperial College COVID-19 response team [20]—see [20]'s Table 2—weighted by the worldwide age distribution [21]. In other words, the IFR for 25–29 year-olds was multiplied by the proportion of the world population aged 25–29, the IFR for 30–34 year-olds was multiplied by the proportion of the world population aged 30–34, and so on, with the sum of the results for all age brackets treated as the world IFR. As best we can determine from the description available, this was the method used by the Imperial College team to estimate fatality rates across countries of different income brackets, but applied to the world rather than only those countries in a specific income bracket. These outcome estimates necessarily remain approximate but were presumed adequate to compare expert and nonexpert predictions.

This study was overseen by the Psychology Research Ethics Committee of the University of Cambridge (approval number PRE.2020.034, amendment 1 April 2020). Participants viewed a Participant Information Sheet and provided written consent via an electronic consent form before accessing the survey.

**Table 2. Proportions of participants from each group (experts, high-numeracy nonexperts, and all nonexperts) for whom the outcome fell within their own 75% confidence intervals.**

| Question no. | Experts | High-numeracy nonexperts | All nonexperts | $X^2$, experts vs. high-numeracy nonexperts (p-value) | $X^2$, experts vs. all nonexperts (p-value) |
|---|---|---|---|---|---|
| 1 | 39/108 (36%) | 78/483 (16%) | 169/1757 (10%) | 22.2 ($p < .001$) | 72.1 ($p < .001$) |
| 2 | 40/100 (40%) | 58/479 (12%) | 133/1737 (8%) | 45.8 ($p < .001$) | 115.9 ($p < .001$) |
| 3 | 41/98 (42%) | 47/466 (10%) | 159/1634 (10%) | 62.0 ($p < .001$) | 93.3 ($p < .001$) |
| 4 | 55/96 (57%) | 129/474 (27%) | 330/1673 (20%) | 33.0 ($p < .001$) | 75.2 ($p < .001$) |

Data and code are available at https://osf.io/dcn5q. See S1 Appendix for questionnaire items.

## Construction of linear opinion pools (consensus distributions)

In addition to calculating the accuracy and calibration of experts and nonexperts as individuals, consensus distributions were generated for each question by aggregating distributions having 75% of the probability mass uniformly distributed within each participant's 75% confidence interval. This required constructing and combining distributions for individual participants, and enabled us to calculate continuous ranked probability scores (CRPS), a common approach to comparing probabilistic forecasts. Although we computed the CRPS of aggregated consensus distributions, if we were to compute it for an individual forecast (after transforming that forecast to a probability distribution in the manner described later in this section), it would most reward participants with relatively narrow 75% confidence intervals that also contained the correct outcome; it would give poorer scores to participants whose 75% confidence intervals contained the correct outcome, but were somewhat wider, and would give especially poor scores to participants with narrow 75% ranges that were not anywhere close to the correct outcome. CRPS is therefore useful in this study as it serves as a measure of accuracy that can be applied to a consensus distribution constructed from participants' 75% confidence intervals, rather than from their point estimates. An alternative, perhaps more straightforward approach would have been to use the weighted interval score [22], which approximates the CRPS and does not require the construction of a full predictive distribution. Constructing consensus distributions also allowed us to better visualize the aggregated predictions.

For each of the four questions, a separate consensus distribution was constructed for experts, nonexperts, and high-numeracy nonexperts, yielding a total of twelve consensus distributions (Fig 1). These distributions assigned zero probability mass to values less than 0 or values greater than a "natural maximum": 1000 for questions 3 & 4, which asked participants to estimate how many people out of every 1000 would experience a particular outcome; and 67 million (the approximate population of the UK) for questions 1 and 2, which asked participants to estimate how many people in the UK would experience a particular outcome. Probabilities were assigned to outcomes between 0 and the natural maximum by averaging together probability distributions constructed for each participant. The distribution for each [expert / high-numeracy nonexpert] was constructed as follows: 75% of the probability mass was distributed uniformly within the interval [*lower*, *upper*], where *lower* and *upper* refer to the lower and upper bounds of the range provided by the participant; the width of this interval is given by *upper*–*lower*. The remaining 25% was distributed uniformly within [*min*, *lower*) ∪ (*upper*, *max*], where *min* was the larger of 0 and *lower*–(*width/2*), and *max* was the smaller of *upper* + *(width/2)* and the natural maximum. This had the effect of restricting the total width of each individual distribution to a width equal to or less than twice the width of the range provided by the participant, and was done in order to ensure that the remaining 25% was allocated to values close to the participant's 75% confidence interval. While the specific choice of how much to restrict the distribution was necessarily somewhat arbitrary, if we were to assume that the participant's 75% confidence interval corresponds to the "middle 75%" (i.e., the 12.5th to 87.5th percentiles) of a Gaussian distribution, two times this width would span from the 1st percentile to the 99th percentile. It therefore seems likely that an individual who is truly 75% confident that a value will fall between *lower* and *upper* would be extremely surprised if the value was less than *lower*–(*width/2*) or higher than *upper* + *(width/2)*. Restricting the distributions within these bounds therefore seemed justified. Finally, probability distributions of different individuals were combined with simple averaging of the probabilities (i.e. 'vertical' combination).

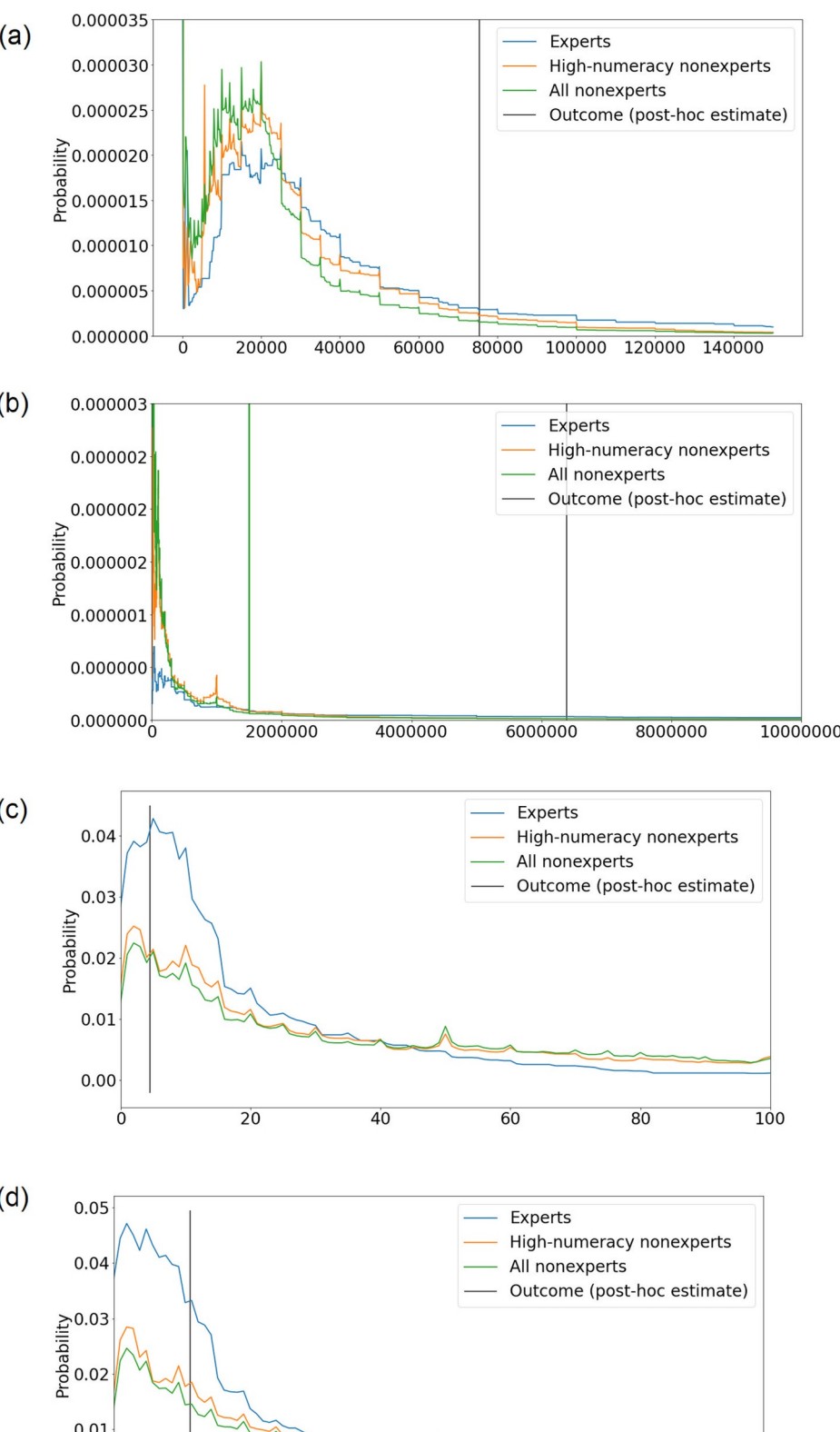

**Fig 1.** Consensus distributions (linear opinion pools) for Q1 (a), Q2 (b), Q3 (c), and Q4 (d). Axes truncated to allow the overall shapes of the distributions to be visible.

For nonparametric distributions, the scoringRules R package [23] can make use of random samples from the forecast distribution to approximate the CRPS. We randomly sampled sets of 5,000 samples from the consensus distributions repeatedly (100 times per distribution), resulting in 100 approximate CRPS scores for each consensus distribution.

## Results

### Accuracy

As reported in Table 1, in terms of absolute error, expert point estimates for each of the four questions were more accurate than those of high-numeracy nonexperts. Mood's median tests indicated that these differences were significant: Q1, $p = .03$, Q2, $p = .04$, Q3, $p < .001$, Q4, $p < .001$. Expert point estimates were also more accurate than those of all nonexperts, Q1, $p < .001$, Q2, $p = .003$, Q3, $p < .001$, Q4, $p < .001$.

Similar results emerged when evaluating relative error rather than absolute error. Several measures of relative error exist; for ease of interpretation we here report the exponential of the absolute value of the log difference measure [24], $\ln(\hat{x} / x)$. This is a natural measure for predictions of this kind as it is scaled to the size of the true outcome. For example, a prediction that is 1/3 the size of the true outcome is treated as having the same amount of relative error as a prediction that is 3 times the size. As shown in Table 1, experts had less relative error than nonexperts, with their medians being lowest for each question. Mood's median tests indicated that differences in relative error between experts and high-numeracy nonexperts were not significant for Q1 ($p = .07$) but were significant at $p < .001$ for the remaining three questions. Differences in relative error between experts and all nonexperts were significant at $p < .001$ for all questions.

### Calibration

Calibration could be assessed for the 93 experts, 1459 nonexperts, and 434 high-numeracy nonexperts who fully answered all questions with clearly interpretable responses. Given that four questions were asked, the number of outcomes falling within an optimally calibrated individual's 75% confidence interval has an expected value of three. Experts came closer to meeting this standard than nonexperts: The median number of outcomes falling within each participant's interval was 2 for experts (mean = 1.82, SD = 1.17), 0 for high-numeracy nonexperts (mean = 0.68, SD = 0.90) and 0 for all nonexperts (mean = 0.49, SD = 0.77). Mood's median tests indicated that the differences in medians between experts and nonexperts, and between experts and high-numeracy nonexperts, were significant, both $p < .001$. 20 of 93 experts (22%), 33 of 1459 nonexperts (2%), and 16 of 434 high-numeracy nonexperts (4%) were calibrated such that exactly 75% (three) of the four outcomes fell within their 75% confidence intervals.

We also calculated the proportion of participants from the given group (experts, high-numeracy nonexperts, or all nonexperts) for whom the outcome ("true outcome estimate") fell within the participant's 75% confidence interval, inclusive; other investigators have treated this proportion as a measure of 'calibration of confidence assessments' for a group [25]. For a group in which each individual was perfectly calibrated, 75% of participants' 75% confidence intervals would be expected to contain the true value. The "true outcome estimate" fell within the 75% confidence intervals for Q1 to Q4 of 36%, 40%, 42%, and 57% of experts respectively,

**Table 3. Descriptive and inferential statistics for the sets of 100 approximate continuous ranked probability scores generated from expert and high-numeracy non-expert consensus distributions.**

| Question no. | Experts | | High-numeracy nonexperts | | Two-tailed t-test comparing experts and high-numeracy nonexperts | | |
|---|---|---|---|---|---|---|---|
| | Mean | SD | Mean | SD | t | df | p |
| 1 | 24,301 | 289 | 31,301 | 364 | 150.66 | 188.27 | < .001 |
| 2 | 3,210,153 | 63,585 | 3,563,702 | 32,721 | 49.44 | 148.00 | < .001 |
| 3 | 7.44 | 0.17 | 26.60 | 0.60 | 306.35 | 115.06 | < .001 |
| 4 | 3.46 | 0.07 | 20.27 | 0.54 | 310.51 | 102.50 | < .001 |

but within the ranges of only 16%, 12%, 10%, and 27% of high-numeracy nonexperts and 10%, 8%, 10%, and 20% of all nonexperts. Chi-squared tests indicated that these proportions differed significantly between experts and high-numeracy nonexperts on every question, and that differences between experts and all nonexperts were likewise significant (Table 2).

## Continuous ranked probability scores

Experts had lower (i.e., better) CRPS than high-numeracy nonexperts on each question, indicating superior forecasting overall. Although it was clear that this was the case from the descriptives alone, two-tailed t-tests were nevertheless used to formally test whether the mean of the 100 approximate CRPS scores calculated from the expert consensus distributions was indeed different than that of those calculated from the high-numeracy nonexpert consensus distributions (Table 3). As the sampling procedure to obtain the inputs for the CRPS analysis was computationally intensive and time-consuming, and as the full nonexpert distributions clearly predicted outcomes as poorly or more poorly than the high-numeracy nonexpert distributions (Fig 1), it was not deemed necessary to repeat this analysis for the full nonexpert consensus distributions.

## Accounting for demographic differences

Experts had a mean age of 42.3 (95% CI 40.0–44.7), slightly younger than nonexperts' mean age of 45.3 (95% CI 44.6–46.1). Experts were also 75% (66% - 83%) male; this proportion was higher than either the high-numeracy nonexpert group—51% (47% - 56%) male—or the group of all nonexperts, 48% (46% - 51%) male. When regressing rank-transformed absolute error on gender, age, and expert/nonexpert status, expert status was significantly associated with lower error for each of the four prediction questions, with betas for expert status being 2 to 16 times the magnitude of the corresponding beta for gender, depending on the question (Table 4). Older age was associated with (lower) error for question 4 only. Similarly, expert status ($\beta = 1.3$) and male gender ($\beta = 0.2$) were both significantly associated with the number of outcomes falling within each participant's 75% confidence interval.

**Table 4. Regressions of calibration and accuracy on gender, age, and expert/nonexpert status.**

| Predictor | β | | | | |
|---|---|---|---|---|---|
| | # outcomes within range | Q1 error | Q2 error | Q3 error | Q4 error |
| Expert status | 1.284*** | -352.22*** | -177.61** | -633.91*** | -588.34*** |
| Age | -0.001 | 0.061 | -0.592 | -0.545 | -1.658* |
| Male gender | 0.162*** | -22.22 | -73.68** | -102.21*** | -82.76** |
| Adjusted $r^2$ | 0.145 | 0.013 | 0.006 | 0.053 | 0.045 |

Note. 'Error' represents rank-transformed absolute error. Stars represent significance at $p < .05$ (*), $p < .01$ (**), $p < .001$ (***).

Numeracy could not be included in these regressions as we did not require experts to complete the Berlin numeracy test. However, when these regressions were repeated but with nonexperts restricted to the subset of high-numeracy nonexperts, expert status continued to significantly predict the number of outcomes falling within each participant's interval, and also continued to significantly predict rank-transformed error for each of the four questions, with the exception of question 2.

Restricting the dataset to only nonexperts allowed numeracy to be included as a predictor along with age and gender. Higher numeracy was significantly associated with more outcomes falling within each participant's 75% confidence range, as well as with lower error on each prediction question. Male gender remained associated with more in-range outcomes, and also remained associated with lower error on three of four prediction questions, but the beta for gender had lower magnitude than the beta for numeracy in each case. As before, age was associated with error for question 4 only.

## Discussion

Despite the limitations of this survey, there are nevertheless a few key lessons to be drawn. First, the experts in our study demonstrated overconfidence: out of the four intervals that experts expected outcomes to fall within 75% of the time, fewer than half of actual outcomes fell within these intervals on average. This is perhaps unsurprising given reports of poor calibration of disease models of COVID-19 and the 2014 Ebola outbreak [8, 12], but it is noteworthy that this was true even when experts were being asked to fill out an informal survey—a context in which most experts presumably did not run their favorite epidemiological model to see what it predicted by year end. In the present case, numbers of deaths and infections by the end of the year were substantially more severe than most expert predictions, unlike in the 2014 Ebola outbreak, when outcomes were less severe than predicted by experts [12]. The common theme seems to be that estimates of the likely intervals in which future observations would fall were too narrow.

Second, nonexpert predictions were less accurate than expert predictions, and nonexperts were more overconfident than experts in their predictions. This was true even of those nonexperts who scored in the top quartile of a standard test of numeracy. Follow-up analyses suggested that these differences were not due to confounds with age or gender. Therefore, although our findings on expert accuracy and overconfidence may read as a cautionary tale against taking expert predictions at face value, it is critical to highlight that we could do worse: we could believe the predictions of people who are *not* experts. We have arguably witnessed many examples of the latter approach being taken by individuals across the globe, sometimes with dire results. As [15] notes, the (in)accuracy of lay predictions is essential context when discussing expert performance. Focusing solely on poor expert performance may simply make nonexperts more adamant about their own preconceptions—not a good thing if they are already even more inaccurate and more overconfident than the experts, as our results suggest.

A key limitation with respect to what this study can tell us about expert predictions in the real world is that there is enormous heterogeneity both among experts and the conditions under which they make forecasts. The concept of an 'expert' as operationalized in this study is extremely broad. It would be especially helpful to understand more about the accuracy of forecasts produced by the subset of experts who are most influential, e.g., those who sit on scientific advisory committees that inform policy. These are a very specific subset of 'experts' who are harder to study, and it is an open question to what degree any findings about off-the-cuff predictions of the 'experts' recruited in our study might generalize to predictions made by those experts who have an explicit mandate to produce forecasts for policymakers.

Furthermore, there are 'experts' who are selected as policy advisors (or as interviewees on media programs) because it is known that they support policies preferred by a particular political party; experts who choose to become involved in policy precisely because they feel more confident than others about what policies should be advocated for; experts who may be incentivized to be more cautious, fearing that reputation and employment opportunities may suffer if they get things too badly wrong; and experts who thrive on media attention (or are even employed as media pundits), who may have more incentive to make attention-grabbing predictions than accurate ones. In other words, individual incentives and incentive structures likely have enormous influence on accuracy and calibration, and it would be a mistake to assume that our results are equally applicable to all subtypes of 'expert'.

However, there are some reasons to believe that the general finding of overconfidence (among both 'experts' and laypeople) is likely to generalize to a number of contexts in which COVID-19 forecasts (and presumably epidemiological forecasts more generally) are made. First, the finding of overconfidence among individuals with relevant subject-matter expertise is consistent with, in the words of Philip Tetlock, "a multi-decade line of psychological research on expert judgment that has shown that experts in a wide range of fields are prone to think they know more than they do (the overconfidence effect)" [26]; other researchers have described overconfidence as "the most ubiquitous bias in studies of calibrated judgments about risks and uncertainties" [27]. Research summarizing relevant studies across a wide variety of fields finds systematic overconfidence in judgments made by both lay predictors and those with relevant expertise [28–30]. This is not solely a laboratory phenomenon. For example, in an analysis of roughly five thousand 90% confidence ranges from 27 studies mostly "performed in the course of applied research in the experts' domains of expertise, not primarily in laboratory studies of overconfidence," and where all respondents were "experienced professionals giving judgments on important real-world problems within their own domains of expertise," Lin and Bier [31] found much variation in the extent of overconfidence, but found overconfidence overall. On average, the percentage of true values falling within respondents' 90% probability intervals was less than 90% for each of the 27 studies they analyzed, although when digging deeper into the questions asked by each study, they found that the level of overconfidence was highly variable and that "some questions even produced under-confidence."

This is not to say that overconfidence in judgments about future events is universal among experts or nonexperts. Overconfidence is influenced by the frequency and kind of feedback that individuals typically receive about their predictions; for example, weather forecasters appear to be particularly well-calibrated [28]. It also seems highly likely that forecasting experience and the incentive structure of the social environment in which the forecasts are made will have an effect. For example, members of the Good Judgment Project, a group of forecasters known for their exceptional performance in a multi-year geopolitical forecasting competition conducted by the U.S. Intelligence Advanced Research Projects Activity, achieved exceptionally high calibration [32]. They attributed this success to the fact that they explicitly aimed to "structure the situation, incentives, and composition" of their team so as to produce accurate and well-calibrated forecasts, and also present evidence that training and interaction in teams was beneficial. We therefore certainly would not conclude that domain experts untrained in forecasting would perform better than large groups of 'nonexperts' who are practiced in forecasting future events. One such group appears to have outperformed experts in infectious disease modelling on multiple COVID-19-related forecasts, on the forecasting platform Metaculus [33]. Promising follow-up research is beginning to combine the predictions of 'nonexpert' forecasters from Metaculus and the Good Judgment Project with those of epidemiological modelers to produce consensus forecasts of hopefully greater accuracy than either in isolation, as well as a 'meta-forecast' which combines this consensus forecast with an ensemble

of forecasts from computational models [34]; the results have yet to be systematically evaluated. Other initiatives to solicit and evaluate a wide range of approaches to epidemiological forecasting, such as the DARPA Chikungunya challenge [35], in combination with research on approaches to aggregating forecasts of subject-matter experts [36] and nonexperts [37–39], have also established promising routes toward improving forecasting of epidemics. In other words, we are not all doomed to be overconfident: there is much that can be done to improve the accuracy and calibration of forecasts, at least in the context of forecasting tournaments.

Forecasting experience and the incentive structure of the social environment are likely to affect predictions made outside of forecasting tournaments as well. For example, Mandel and Barnes' analysis [40] of over 3,500 geopolitical forecasts from intelligence analysts, extracted from real-world reports, found systematic underconfidence rather than the overconfidence seen in Tetlock's famous studies of expert geopolitical forecasting, or the excellent calibration seen in forecasting tournaments. Although there were many differences that may have contributed to the different findings, the researchers noted one plausible factor was that "accountability pressures on analysts are far greater than those placed on forecasters in geopolitical tournaments," citing research from Tetlock & Kim showing improved calibration in the presence of social accountability [41]. Incentives to be accurate and well-calibrated in our study were low for experts and nonexperts alike. It is possible that, with respect to public predictions which may influence individual behavior, some experts may have an especially strong incentive to be as correct and cautious as possible, feeling that their reputations are on the line and that their predictions will be subject to future scrutiny. Intuitively, this seems likeliest to be true for experts who feel that they benefit much more from a reputation for accuracy than from media attention—although these experts may also be least likely to voice public predictions about highly uncertain events in the first place.

Unfortunately, however, for many public predictions, there are reasons to believe the incentive structure may be the other way round. For example, Tetlock and colleagues asked expert participants "how often they advised policy makers, consulted with government or business, and were solicited by the media for interviews", reporting a significant positive correlation between these assessments and the degree of overconfidence exhibited by these experts in geopolitical predictions [42]. They also reported a positive correlation between overconfidence and Google search counts (used as a proxy for the number of times participants were mentioned in the media). Tetlock points out that the causal links may be bidirectional: "On one hand, overconfident experts may be more quotable and attract more media attention. On the other, overconfident experts may also be more likely to seek out the attention" ([42], p. 63). Either way, the result is the same: predictions made by overconfident experts may be the most visible.

In sum, our findings of overconfidence in lay and expert COVID-19 predictions are consistent with what would be expected from literature on predictions and judgments in other domains. They seem likely to generalize to common real-world contexts in which everyday people encounter expert or nonexpert predictions (e.g., within traditional or social media), but not necessarily to all such contexts. They also may not generalize to contexts where accuracy and good calibration are strongly incentivized, or in which individuals receive systematic training or practice in forecasting with regular feedback.

Are there other data suggesting overconfidence in experts or nonexperts of COVID-19 case count and death prediction? There exist a very interesting set of reports associated with the study mentioned in the introduction by McAndrew and Reich, which solicited COVID-19-related predictions from U.S. experts in a total of twelve surveys. The fifth survey (administered March 16–17, 2020), the first to ask both about the 'smallest, most likely, and largest' number of U.S. COVID-19 deaths they expected by the end of 2020, as well as about the

'smallest, most likely, and largest' number of U.S. COVID-19 cases that they expected to have been reported for specific dates in the future, was completed by 18 experts in infectious disease modelling [43]. Although the data for individual expert predictions is not available, the 80% confidence interval of the experts' consensus distribution for reported U.S. COVID-19 cases by March 22$^{nd}$ was reported as 7,061–24,180 [43], and the 80% confidence interval for March 29$^{th}$ was 10,500–81,500, according to a news article appearing on ABC News' FiveThirtyEight about the same survey [44]. The 'smallest, most likely, and largest' estimates of each expert for the latter prediction were also visualized in the article. Ultimately, according to the 'truth database' later compiled by the survey authors [45], the true outcomes for the corresponding questions were 33,404 and 139,061, respectively, both well outside the 80% confidence intervals. From FiveThirtyEight's visualization, it can be inferred that the true March 29$^{th}$ count exceeded the high-end estimate of 15 of the 18 modellers. On the other hand, the expert consensus 80% confidence interval for the number of U.S. COVID-19 deaths in 2020 was 195,000–1.2 million, with a point estimate of 195,000; the CDC's ultimate 2020 COVID-19 death count of 379,705 [46] fell within this range, although it exceeded the high-end estimate of 240,000 announced by U.S. administration scientists on 31 March 2020 [47]. By the time of McAndrew and Reich's final survey, administered May 4–5, they were also soliciting crowd-sourced predictions on the ultimate number of U.S. COVID-19 deaths expected by the end of 2020 from Metaculus; by this time, experts were predicting a median 256,000 deaths (80% CI 118,000–1.2 million), and Metaculus predictors a median 197,000 (80% CI 120,000–460,000) [48].

It is difficult to draw too many conclusions from these examples, but they do provide an example of overconfident COVID-19 predictions even among infectious disease modelers, making predictions for just one week into the future, early in the pandemic. The non-linear dynamic nature of infectious diseases makes possible futures especially uncertain–small initial differences in infection parameters can lead to big differences in outcomes with time–and it certainly seems plausible that this makes it challenging to estimate one's own level of certainty. However, overconfidence is not necessarily restricted to experts or to epidemiological forecasts, as the prior literature we have discussed makes clear.

We also expect that our finding of poorer accuracy in COVID-19-related predictions for the "person on the street" versus people with relevant subject-matter expertise is likely to generalize to some degree, but not to all nonexperts. In addition to being a common-sensical finding, 'relevant subject-matter expertise' typically includes knowledge of what real-world data sources contain the most reliable information; for example, it seems plausible that our 'experts' did well on questions 3 and 4 simply because they were aware of preliminary estimates of the infection fatality rate, whereas our nonexperts may not have been. Individuals with professional backgrounds in statistics, mathematical modelling, and epidemiology also seem likelier to have had a greater familiarity with the raw data around COVID-19 infections and death rates, a firmer grounding in how to interpret that data, and the high levels of uncertainty associated with epidemiological forecasts in general. As mentioned, however, the impressive performance of practiced forecasters on crowdsourcing platforms suggests that domain expertise is not necessarily a prerequisite for good forecasting.

Other limitations of this study include convenience sampling of experts and a small number of correlated questions. Participants were also asked to produce point estimates before they were asked to produce ranges, which is known to anchor responses toward the point estimate [49], so participants may have seemed more overconfident than they would have with other elicitation methods. Nevertheless, given the stark differences between expert and nonexpert accuracy and calibration levels, it seems unlikely that alternate elicitation methods would erase these differences. Additionally, we purposefully cast a broad net in terms of our recruitment of

experts, and it is certainly possible that a more rigorous process for selecting individuals with the most relevant subject-matter expertise would have resulted in a set of experts who made more accurate predictions. However, a more careful selection of experts would presumably have simply made the differences between nonexpert and expert performance even more stark (if it had any effect at all), rather than the reverse, so there seems little reason to question the performance differences between experts and nonexperts.

## Conclusions

Much of the discussion around communicating forecasts in the COVID-19 pandemic has centered around tradeoffs in communicating uncertainty with respect to public trust. For example, in some contexts downplaying uncertainties may shore up public trust in the short term, but confident predictions that later turn out to be wrong may reduce public trust in science [50]. Overall, our results underscore a need for individuals with expertise in fields relevant to forecasting epidemiological outcomes (and who communicate about these forecasts publicly) to consider broad ranges of possible outcomes as plausible, and to consider communicating this high level of uncertainty to nonexperts. The ultimate message may be that "the experts have much to learn, but they also have much to teach" [15].

Given the continued impact of COVID-19 and risks of other future pandemics, further research into improved epidemiological forecasting may prove vital. In the meantime, we must all learn to acknowledge and admit that the uncertainties may be greater than we think they are, whether we are experts or not.

## Supporting information

**S1 Appendix. Questionnaire items.** Expert and nonexpert questionnaire items.
(DOCX)

## Acknowledgments

We would like to thank all participants and administrators who made this work possible.

## Author Contributions

**Conceptualization:** Gabriel Recchia, Alexandra L. J. Freeman.

**Data curation:** Gabriel Recchia.

**Formal analysis:** Gabriel Recchia.

**Investigation:** Gabriel Recchia.

**Methodology:** Gabriel Recchia, Alexandra L. J. Freeman, David Spiegelhalter.

**Supervision:** Alexandra L. J. Freeman.

**Visualization:** Gabriel Recchia.

**Writing – review & editing:** Alexandra L. J. Freeman, David Spiegelhalter.

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
