## [Decision Letter · Decision Letter 0]

20 Mar 2021

PONE-D-21-04714

How well did experts and laypeople forecast the size of the COVID-19 pandemic?

PLOS ONE

Dear Dr. Recchia,

Thank you for submitting your manuscript to PLOS ONE. After careful consideration, we feel that it has merit but does not fully meet PLOS ONE’s publication criteria as it currently stands. Therefore, we invite you to submit a revised version of the manuscript that addresses the points raised during the review process.

We look forward to receiving your revised manuscript.

Kind regards,

Martial L Ndeffo Mbah, Ph.D

Academic Editor

PLOS ONE

Additional Editor Comments:

This is a very interesting manuscript addressing a timely question of disease forecasting. Though reviewers mainly pointed out some minor issues that need to be addressed, providing more clarification on the use of terms 'expert' and 'non-expert' will greatly improve the quality and potential impact of the study. To this effect, I fully agree with Reviewer #1 suggestions, and strongly encourage you to address them thoroughly.

Journal Requirements:

Reviewers' comments:

Reviewer's Responses to Questions

**Comments to the Author**

1. Is the manuscript technically sound, and do the data support the conclusions?

Reviewer #1: Partly

Reviewer #2: Yes

2. Has the statistical analysis been performed appropriately and rigorously? 

Reviewer #1: Yes

Reviewer #2: No

3. Have the authors made all data underlying the findings in their manuscript fully available?

Reviewer #1: Yes

Reviewer #2: Yes

4. Is the manuscript presented in an intelligible fashion and written in standard English?

Reviewer #1: Yes

Reviewer #2: Yes

5. Review Comments to the Author

Reviewer #1: Summary

While expert opinions have exerted a large influence on public policy during the Covid-19 pandemic, relatively little research has been done that examines the accuracy of such expert predictions.

To fill that gap, the paper shows the results of a survey done in April on experts and non-experts who were asked to submit a prediction for four different covid-related targets. The paper makes a compelling case that experts (and to an even greater extent non-experts) have a tendency to be overly certain in their estimates and underestimate the possibility of extreme events. It is interesting to see that this tendency was different between experts and non-experts. It is also interesting that off-the-cuff predictions, on average, probably are not helpful to inform public policy.

Major issues

The idea of comparing experts and non-experts is in principle very interesting - especially given the importance of expert opinion for shaping public policy. One factor that makes studying this subject difficult, however, is the immense variability hidden behind the terms 'expert' and 'non-expert' that is (naturally) hard to capture. For example, a scientific policy advisor may be a quite different type of expert (with potentially much better forecasting skills) than someone who simply has a statistics background. Similarly, a layperson putting a lot amount of effort into forecasting questions on Metaculus, or someone who is a Superforecaster (see https://goodjudgment.com/) may make forecasts very differently than the average person who would participate in a study through Prolific Academic or Respondi.

One way in which I especially fear study participants may differ from either 'experts' who inform policy or voice predictions publicly, or 'non-experts' who make public predictions is 'skin in the game' and the incentive structure they are faced with. Experts who make public predictions and who take responsibility may have very strong incentives to be as correct as possible (and possibly as cautious). Even if they make off-the-cuff prediction, they will probably be aware of public scrutiny. Similarly, laypeople who predict to earn money or to rank highly on a public leaderboard also have an incentive to be right. This likely greatly enhances predictive accuracy (see e.g. literature on Superforecasters). Experts surveyed in the study however, as I understood, had no real incentive to be right (except for the fact that they were recruited as 'experts' and therefore possibly felt some psychological pressure to live up to that standard). Non-experts presumably had no incentive at all to be right and maybe even had a financial incentive (in terms of their hourly rate) to complete the online survey quickly. The authors briefly mention how some people on Metaculus may be outperforming experts consistently, but unfortunately do not address this further or contextualise it in light of their research.

I feel it would be helpful if the authors were more explicit on these limitations or address them otherwise. They do state unequivocally how they recruited their sample of 'experts' and 'non-experts', but could be clearer on the implications of that selection process and the study design. It would help to expand on how comparable the 'experts' in the study are to the 'experts' that actually do inform public policy. In the paper the same term 'expert' is used in two ways: First in the introduction and conclusion that talks about how important expert opinion is for public policy, and secondly to denote the group of people in the study with a background in certain fields. To the reader, this implies some comparability. I think it needs more explanation to argue that forecasts from the former group are actually comparable to predictions from the latter - or an explicit statement that they may not be. One option could be to call them 'study participants with a background in subject relevant fields' (or probably a catchier version of that) in the paper. Similarly, I think it should be addressed that the group of laypeople in the study may not be able to represent all kinds of 'non-experts' predictions, or even the ones we might care about most (e.g. Metaculus, Predictit etc.). That being said I do find it intuitively plausible that all experts, including those making public predictions have a tendency to be overconfident, even if they are incentivised to be right. I just believe this link needs more supporting evidence. One way to do show this could be to check for explicit predictions made by experts around the same time and see whether they are comparable with what the study participants predicted.

Minor issues

. line 66 page 3 the reporting of the Ebola study is worded slightly confusingly. What is 4 of 7?

- line 123 I believe it should be 'divided by 1,000' instead of multiplied by 1,000

- In general, I feel it is somewhat unfortunate that for 3 out of 4 questions, there is no actual "truth data", but instead only estimates. This should ideally be thought of beforehand for future studies

- Specifically for question 3 (lines 123ff) it is unclear to me what was multiplied with what to obtain death numbers. I'm also not quite clear why the authors did not simply take publicly reported death numbers?

- Regarding the construction of linear opinion pools: the authors could maybe be clearer on

a) why they chose to construct distributions for every individual expert at all. Was this just for the purpose of scoring, or was this mainly to construct an overall aggregate distribution?

b) how their choice of restricting the distribution with 'width/2' was motivated?

c) how exactly they aggregated individual expert distributions to one distribution. Did they combine the CDFs horizontally, i.e. use a quantile average, or vertically, i.e. construct a mixture distribution?

- Regarding the scoring of forecasts it could be made clearer why the authors used the crps instead of weighted interval score (e.g. Bracher et al. 2021 https://doi.org/10.1371/journal.pcbi.1008618). It seems much more intuitive to calculate the wis for every individual expert and average it for every subgroup (experts, non-experts etc) the authors are interested. This is much easier than constructing a full predictive distribution and using the crps to score it. A software implementation of the WIS is for example available in the scoringutils package (https://github.com/epiforecasts/scoringutils - Disclaimer: I'm an author of that package).

- Regarding the assessment in terms of relative scores in lines 176ff: I am unsure whether the relative assessment is actually necessary, as the main message is already clear in terms of absolute differences and so a relative assessment doesn't seem to add much.

- stylewise, the reporting of the numerical results in the sections "Calibration" and "Continuos ranked probabilty" is somewhat hard to follow as there are a lot of numbers, maybe a table would be better suited to present the different p-values and tests performed? Especially in the section on the crps, the authors could be clearer on what numbers they report (for example, I'm not sure what the negative numbers mean).

- A minor point regarding the assessment of calibation in lines 188ff: From the text I understand that the authors assume that a forecaster whose prediction intervals cover more of the true values are better calibrated. While this seems true in the case where all participants are overly confident in their predictions, it doesn't hold in general: a forecaster who covers 100% of the observed values with his prediction intervals (instead of the 75% aimed for) is not necessarily better calibrated than one who is right 50% of the time.

Reviewer #2: The paper analyzes the precision of forecast regarding the magnitude of the COVID-19 pandemic by experts and non-experts. They compare the forecast in spring 2020 at the initial stage of the pandemic with the outcomes in December in the same year. They found that experts perform better in both accuracy of prediction and accuracy of levels of confidence than non-experts. The research addresses an important policy-relevant question.

Major comments.

1. They compare raw data across groups. However, individual characteristics, such as gender or age, may explain some of the observed difference. Please compare the distribution (or mean) of individual characteristics across the three groups. I suspect the distribution is not necessarily similar given their sampling scheme. If it is the case, is there any way to control for the difference in some of the analyses?

Comments.

2. P4. L86.There are two ways to survey non-experts: Prolific Academic and Respondi.com. What’s the reason for using two ways for survey? Is there any difference in respondents’ characteristics and/or responses? Please explain.

3. P4. L84. How do they pick up the particular value of 75% for calibration? Is this the common number used in literature?

4. P9. L162. Please explain what CRPS shows in the method. It seems it analyzes the accuracy of forecasting, but I would like to know more details on why these analyses are necessary.

5. P10. L211. Please explain for what hypothesis test the t statistics are used in CRPS analyses.

6. Discussion. The next obvious question is where the difference between experts and non-experts come from. Is it because of their scientific knowledge, difference in information source or something else? I understand this is not the question the current study could address from the survey, but any discussion would be informative for readers.

7. Discussion. Does this study have any implication for the context outside of non-infectious diseases? In other words, to what extent this is related to the non-linear dynamic nature of infectious diseases?

6. PLOS authors have the option to publish the peer review history of their article (what does this mean?). If published, this will include your full peer review and any attached files.

Reviewer #1: **Yes: **Nikos I. Bosse

Reviewer #2: No

---

## [Author Response · Author response to Decision Letter 0]

1 Apr 2021

Please find our response to reviewers' comments at the end of the full submitted PDF.

---

## [Decision Letter · Decision Letter 1]

19 Apr 2021

How well did experts and laypeople forecast the size of the COVID-19 pandemic?

PONE-D-21-04714R1

Dear Dr. Recchia,

We’re pleased to inform you that your manuscript has been judged scientifically suitable for publication and will be formally accepted for publication once it meets all outstanding technical requirements.

Kind regards,

Martial L Ndeffo Mbah, Ph.D

Academic Editor

PLOS ONE

Additional Editor Comments (optional):

Reviewers' comments:

Reviewer's Responses to Questions

**Comments to the Author**

1. If the authors have adequately addressed your comments raised in a previous round of review and you feel that this manuscript is now acceptable for publication, you may indicate that here to bypass the “Comments to the Author” section, enter your conflict of interest statement in the “Confidential to Editor” section, and submit your "Accept" recommendation.

Reviewer #1: (No Response)

Reviewer #2: All comments have been addressed

2. Is the manuscript technically sound, and do the data support the conclusions?

Reviewer #1: Yes

Reviewer #2: Yes

3. Has the statistical analysis been performed appropriately and rigorously? 

Reviewer #1: Yes

Reviewer #2: Yes

4. Have the authors made all data underlying the findings in their manuscript fully available?

Reviewer #1: Yes

Reviewer #2: Yes

5. Is the manuscript presented in an intelligible fashion and written in standard English?

Reviewer #1: Yes

Reviewer #2: Yes

6. Review Comments to the Author

Reviewer #1: Thank you for addressing our comments in a very thorough manner. I feel the paper is much improved and am looking forward to seeing it published. Especially the discussion I found very good and helpful.

I only have a few very minor notes about the paragraph on CRPS and the consensus distribution:

- I think your explanation of the behaviour of the CRPS is very good and valuable. Maybe adding the word "precision" to the sentence "... serves as a measure of accuracy that can be applied to a consensus distribution constructed" would be a good idea, as the CRPS assesses sharpness subject to calibration (see Gneiting et al. 2007, Probabilistic forecasts, calibration and sharpness)

- Explaining the CRPS in the context of one individual forecast is maybe slightly misleading, because that is exactly the scenario where the WIS should be used instead of the CRPS. If you have a limited set of quantiles, than the WIS is the better way to approximate CRPS than to assume an arbitrary distribution and calculate the CRPS based on that distribution.

- In terms of the aggregate distribution I still find it hard to judge what really happens when you aggregate using assumed distributions instead of the WIS. I therefore still believe that using the WIS would be cleaner and simpler in terms of the statistical analysis. But I don't believe that this would change results in a meaningful way. Out of personal curiosity I played around with the data you provided and wrote some example code that gave striking results in the same direction you found. Please feel free to make use of this if you like, but also feel free to just ignore it. I think conclusions in the paper are fine without and that this shouldn't block publication.

```R

library(data.table)

install_github("epiforecasts/scoringutils")

library(scoringutils)

#read in data for experts and do same data manipulation to look at one of the targets

# get a quantile column, rename as 'true_value' and 'prediction'

data <- data.table::fread("uk_experts.csv")

data <- data[, .(country_death_estimate, country_death_lower, country_death_upper)]

data[, id := 1:nrow(data)]

data <- melt(data, id.vars = "id")

data[, true_value := 75346]

data[variable == "country_death_estimate", quantile := 0.5]

data[variable == "country_death_upper", quantile := 0.75]

data[variable == "country_death_lower", quantile := 0.25]

data[, variable := NULL]

data[, model := "experts"]

#read in data for high numeracy group and do same data manipulation to look at one of the targets

data1 <- data.table::fread("uk_april_high_numeracy.csv")

data1 <- data1[, .(country_death_estimate, country_death_lower, country_death_upper)]

data1[, id := 1:nrow(data1)]

data1 <- melt(data1, id.vars = "id")

data1[, true_value := 75346]

data1[variable == "country_death_estimate", quantile := 0.5]

data1[variable == "country_death_upper", quantile := 0.75]

data1[variable == "country_death_lower", quantile := 0.25]

data1[, variable := NULL]

data1[, model := "high_num"]

data <- rbindlist(list(data, data1))

setnames(data, old = "value", new = "prediction")

res <- eval_forecasts(data, summarise_by = "model")

res

wis_components(res) #could also be faceted for the different targets

```

Reviewer #2: The authors have adequately responded to my previous comments, and I recommend the current version of the manuscript will be published in the journal.

7. PLOS authors have the option to publish the peer review history of their article (what does this mean?). If published, this will include your full peer review and any attached files.

Reviewer #1: **Yes: **Nikos I. Bosse

Reviewer #2: No

---

## [Editor Report · Acceptance letter]

28 Apr 2021

PONE-D-21-04714R1 

How well did experts and laypeople forecast the size of the COVID-19 pandemic? 

Dear Dr. Recchia:

I'm pleased to inform you that your manuscript has been deemed suitable for publication in PLOS ONE. Congratulations! Your manuscript is now with our production department. 

Kind regards, 

on behalf of

Dr. Martial L Ndeffo Mbah 

Academic Editor

PLOS ONE